# DENV-specific IgA contributes protective and non-pathologic function during antibody-dependent enhancement of DENV infection

**Adam D. Wegman[1], Mitchell J. Waldran[1], Lauren E. Bahr[1], Joseph Q. Lu[1,2], Kristen E. Baxter[2], Stephen J. Thomas[1,2], Adam T. Waickman[1,2]***

**1** Department of Microbiology and Immunology, State University of New York Upstate Medical University, Syracuse, New York, United States of America, **2** Institute for Global Health and Translational Sciences, State University of New York Upstate Medical University, Syracuse, New York, United States of America

\* waickmaa@upstate.edu

## Abstract

Dengue represents a growing public health burden worldwide, accounting for approximately 100 million symptomatic cases and tens of thousands of fatalities yearly. Prior infection with one serotype of dengue virus (DENV) is the greatest known risk factor for severe disease upon secondary infection with a heterologous serotype, a risk which increases as serotypes co-circulate in endemic regions. This disease risk is thought to be mediated by IgG-isotype antibodies raised during a primary infection, which poorly neutralize heterologous DENV serotypes and instead opsonize virions for uptake by FcγR-bearing cells. This antibody-dependent enhancement (ADE) of infection leads to a larger proportion of susceptible cells infected, higher viremia and greater immunopathology. We have previously characterized the induction of a serum IgA response, along with the typical IgM and IgG responses, during dengue infection, and have shown that DENV-reactive IgA can neutralize DENV and competitively antagonize IgG-mediated ADE. Here, we evaluate the potential for IgA itself to cause ADE. We show that IgG, but not IgA, mediated ADE of infection in cells expressing both FcαR and FcγRs. IgG-mediated ADE stimulated significantly higher pro-inflammatory cytokine production by primary human macrophages, while IgA did not affect, or slightly suppressed, this production. Mechanistically, we show that DENV/IgG immune complexes bind susceptible cells significantly more efficiently than DENV/IgA complexes or virus alone. Finally, we show that over the course of primary dengue infection, the expression of FcγRI (CD64) increases during the period of acute viremia, while FcγRIIa (CD32) and FcαR (CD89) expression decreases, thereby further limiting the ability of IgA to facilitate ADE in the presence of DENV. Overall, these data illustrate the distinct protective role of IgA during ADE of dengue infection and highlight the potential therapeutic and prognostic value of DENV-specific IgA.

## Author summary

Dengue virus is a widespread and growing public health challenge, causing an estimated 100 million symptomatic infections every year. A unique feature of dengue is that sub-

**Data Availability Statement:** The authors declare that all data supporting the findings of this study

are available in the data file provided with this article or by contacting the SUNY Upstate Medical University department of microbiology and immunology at biosci@upstate.edu.

**Funding:** Funding for this research was provided by the State of New York (ATW) and the Merck Investigator Studies Program (ATW). The funders had no role in study design, data collection and analysis, decision to publish, or preparation of the manuscript.

**Competing interests:** ADW and ATW are co-inventors on the provisional patent "IgA monoclonal antibodies as a prophylactic and therapeutic treatment for acute flavivirus infection." ATW and SJT are co-founders of Azimuth Biologics, Inc. The remaining authors declare that the research was conducted in the absence of any commercial or financial relationships that could be construed as a potential conflict of interest.

neutralizing levels of virion-specific antibodies generated by a prior infection can increase the severity of a subsequent infection through a mechanism known as antibody-dependent enhancement (ADE). This is thought to be mediated by IgG isotype antibodies that bind the virus and facilitate its uptake via antibody receptors expressed on immune cells. However, IgG is not the only antibody isotype elicited by dengue virus. Our group and others have described a significant IgA response following dengue infection, especially following primary infections and mild secondary infections. It is currently unclear how dengue virus specific IgA contributes to the anti-dengue immune response. Our results here demonstrate that IgA is incapable of facilitating ADE due to the low affinity of IgA for its cognate Fc receptor. This positions IgA as a potential biomarker of dengue severity and highlights the potential therapeutic value of DENV-specific IgA.

## Introduction

Dengue virus (DENV) is a prevalent arboviral pathogen, transmitted primarily by the tropical and subtropical mosquito species *Aedes aegypti* and *Ae. Albopictus*. Dengue represents a major global disease burden: at least half of the world's population is at risk of infection, 100 million of whom develop symptomatic disease per year, leading to at least 20,000 fatalities [1,2]. There are no specific antiviral therapeutics for dengue, and the only US-FDA licensed vaccine as of August 2023 is restricted in its indication to seropositive adults [3]. Clinically, ~75% of DENV infections are asymptomatic, while uncomplicated dengue present as a sudden-onset fever, often accompanied by myalgias, arthralgias, retro-orbital headache, nausea, and rash, and may include epistaxis, petechiae, and minor gingival bleeding [2,4]. However, a small percentage enter a "critical phase" of illness, characterized by increased vascular permeability and plasma leakage or bleeding. The cause of progression to severe dengue is incompletely understood, and no unifying mechanism has been described [5–9].

The DENVs comprise four immunologically and genetically distinct serotypes [10–13]. Following primary infection, the antibody response to the infecting serotype is thought to durably protect against reinfection with a homologous serotype, but cross-protection against infection by heterologous serotypes wanes after 6 months to 2 years [14]. Following this interval of effective cross-protection, secondary infection with a heterologous serotype represents the greatest known risk factor for developing severe disease [15]. One proposed mechanistic contributor to this phenomenon is antibody-dependent enhancement (ADE) of infection, wherein a sub-neutralizing pool of IgG isotype antibodies raised against a primary infection fails to neutralize the secondary infection and instead opsonizes the virions for increased uptake by permissive FcγR-bearing cells [16,17]. Multiple lines of evidence have supported this notion: epidemiologically, with increased severe disease in DENV-experienced populations during a subsequent outbreak of a different serotype; clinically, with increased severe disease in infants born to dengue-immune mothers; and experimentally, with increased peak viremia in nonhuman primates passively immunized DENV immune sera or monoclonal antibodies [14,18–22]. Accordingly, significant effort has been put into defining the serologic profiles elicited by both infection and vaccination, with multiple putative correlates of risk and protection described.

The literature on dengue serology has focused almost exclusively on IgM and IgG isotype antibodies, for several reasons. One of the first papers describing ADE as a phenomenon compared the enhancement potential of IgM and IgG, but not IgA isotype antibodies, and concluded that "the infection-enhancing factor was a noncytophilic antibody of the IgG class" [16]. Later, the IgM/IgG ratio became a routine measurement used to differentiate primary

from secondary infections [23], further contributing to the focus on these two isotypes. However, in addition to IgM and IgG, there is a serum IgA component to the antibody response. There is a small body of literature examining this, including initial attempts to correlate isotypes and subclasses with clinical outcome [24], as well as evaluation of serum IgA as a potential diagnostic tool [25]. Our lab and colleagues have characterized the IgA response at the serological and single-cell (plasmablast) levels during dengue infection [26,27]. We have previously shown that DENV-reactive monoclonal IgA can antagonize IgG-mediated ADE in a dose-dependent manner in the FcγR-bearing K562 cell line [28]. Since ADE occurs via phagocytosis [29], and the FcαR is capable of mediating phagocytosis of IgA-opsonized targets [30–33], the capacity of IgA to contribute to ADE of DENV infection remains unclear.

Here, we demonstrate that IgG, but not IgA, mediates ADE of DENV infection in cell lines and primary human cells which express of FcγR and FcαR. IgG-mediated ADE—but not IgA—increases production of viral particles after infection as well as the secretion of pro-inflammatory cytokines by primary human macrophages. Mechanistically, we show that DENV/IgG immune complexes bind more effectively to FcγR/FcαR expressing cells than DENV/IgA immune complexes, indicating that the inability of IgA to mediate ADE is largely due to the low affinity of IgA to its cognate Fc receptor. Finally, we highlight the dynamic nature of FcγRs and FcαR expressing during acute primary DENV infection, wherein FcγR expression is dramatically increased and FcαR expressing is suppressed. In their totality these results illustrate the distinct protective role of IgA during ADE of dengue infection and highlight the potential therapeutic and prognostic value of DENV-specific IgA during acute dengue.

## Results

### IgG, but not IgA, mediates ADE in FcγR/FcαR expressing cells

Our group has previously demonstrated that DENV-specific IgA antagonizes IgG-mediated antibody-dependent enhancement of DENV infection in the FcγR-bearing K562 cell line [28]. However, while the K562 line is routinely used for *in vitro* ADE assays, it does not express the human myeloid-restricted FcαR (CD89) [34]. CD89 is broadly expressed by monocytes, macrophages and dendritic cells and can facilitate antibody-dependent phagocytosis of immune complexes [30,31]. Therefore, our previous work left unaddressed the potential contribution of DENV-specific IgA to ADE via FcαR. To fill this knowledge gap, we utilized the U937 pro-monocytic cell line, which is routinely used in studies of DENV ADE and has been described to express high levels of FcαR [18,35,36]. We first confirmed the expression of the Fc-receptors and observed that the U937 cell line expresses robust levels of both FcγRIIa (CD32), FcγRI (CD64), and FcαR (CD89), but little or no surface DC-SIGN (CD209) (Fig 1A). To assess the ability of DENV-specific IgG and IgA to enhance DENV infection in U937 cells we utilized a previously described DENV-specific monoclonal antibody (VDB33) which we synthesized with either an IgG or IgA Fc domain [26,28]. The conversion of this parentally IgG1 isotype antibody to an IgA1 format did not impact the binding or neutralization activity of the antibody (S1 Fig) [26,28]. DENV/IgG or DENV/IgA immune complexes were formed with a fixed amount of virus and a variable amount of IgG or IgA and then added to the U937 cells and the frequency of DENV infected cells determined by flow cytometry after 48 hours. Consistent with previously published results, our DENV-specific IgG mAb potently enhanced DENV infection in U937 cells over the infection achieved in the absence of antibody in a titratable fashion (Figs 1B, 1C, S2 and S3). Consistent with other previously-published reports [37], both FcγRIIa and FcγRI contributed to the IgG-mediated infection-enhancement in this model (S4 Fig). However, despite expressing FcαR, no enhancement of DENV infection was observed in U937 cells cultured with IgA/DENV immune complexes.

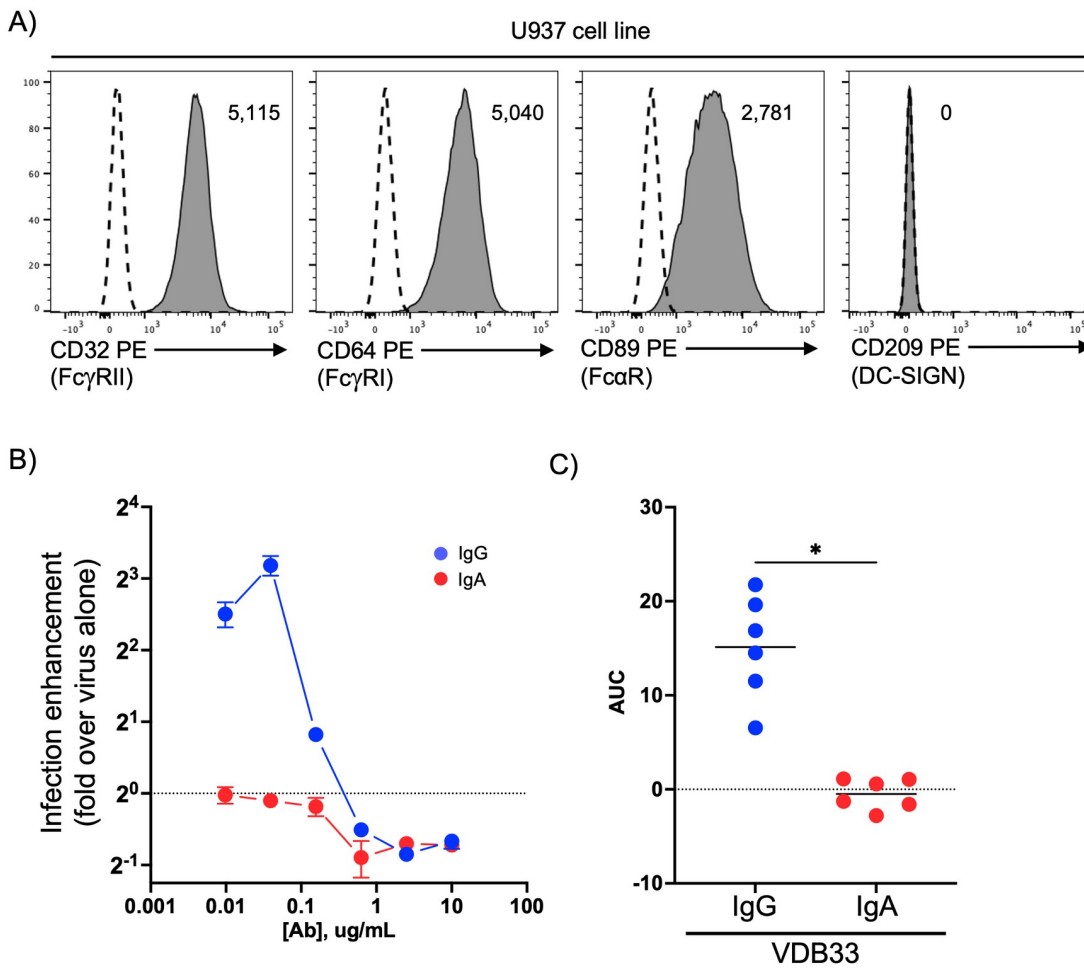

**Fig 1. DENV-specific IgG, but not IgA, mediates ADE in U937 cells. A)** Quantification of FcγRIIa (CD32), FcγRI (CD64), FcαR (CD89), and DC-SIGN (CD209) expression on U937 cells by flow cytometry. Histogram labels are isotype control-subtracted geometric mean fluorescence intensity (MFI) values. Dashed line indicates isotype control staining. **B)** Assessment of DENV infection-enhancing activity of DENV-specific IgG and IgA in U937 cells. Data shown as fold change in U937 cell infection frequency relative to DENV alone at the indicated antibody concentration. Dashed line indicates infection rate observed with virus alone, set to a value of 1 for each biological replicate. **C)** The area under the ADE curves for each of 6 independent biological replicates of IgG and IgA infection experiments relative to infection achieved with DENV alone. * $p < 0.05$, Wilcoxon matched-pairs test.

## IgG, but not IgA, mediates ADE in human macrophages

While the U937 cell line is a widely used and tractable system for ADE studies it is not a fully representative target cell for DENV. As a myeloleukemic cell line, it is replicatively immortal, whereas DENV infects non-replicating cells of the myeloid lineage [38,39]. Moreover, U937 cells are immature pro-monocytes, and Fc-receptor expression levels are known to be modulated during the normal lifespan of monocytes as well as during their differentiation into macrophages [40].

To address this limitation, we next utilized human monocyte-derived macrophages as a target of DENV ADE [37,41]. Consistent with previous reports and *ex vivo* phenotyping [37], our M-CSF differentiated monocytes express high levels of CD14 and CD163 and retained expression of FcγRIIa, FcγRI, and FcαR (Figs 2A, 2B and S5). Importantly, our monocyte differentiation protocol did not induce expression of the dendritic cell-specific CD209, further

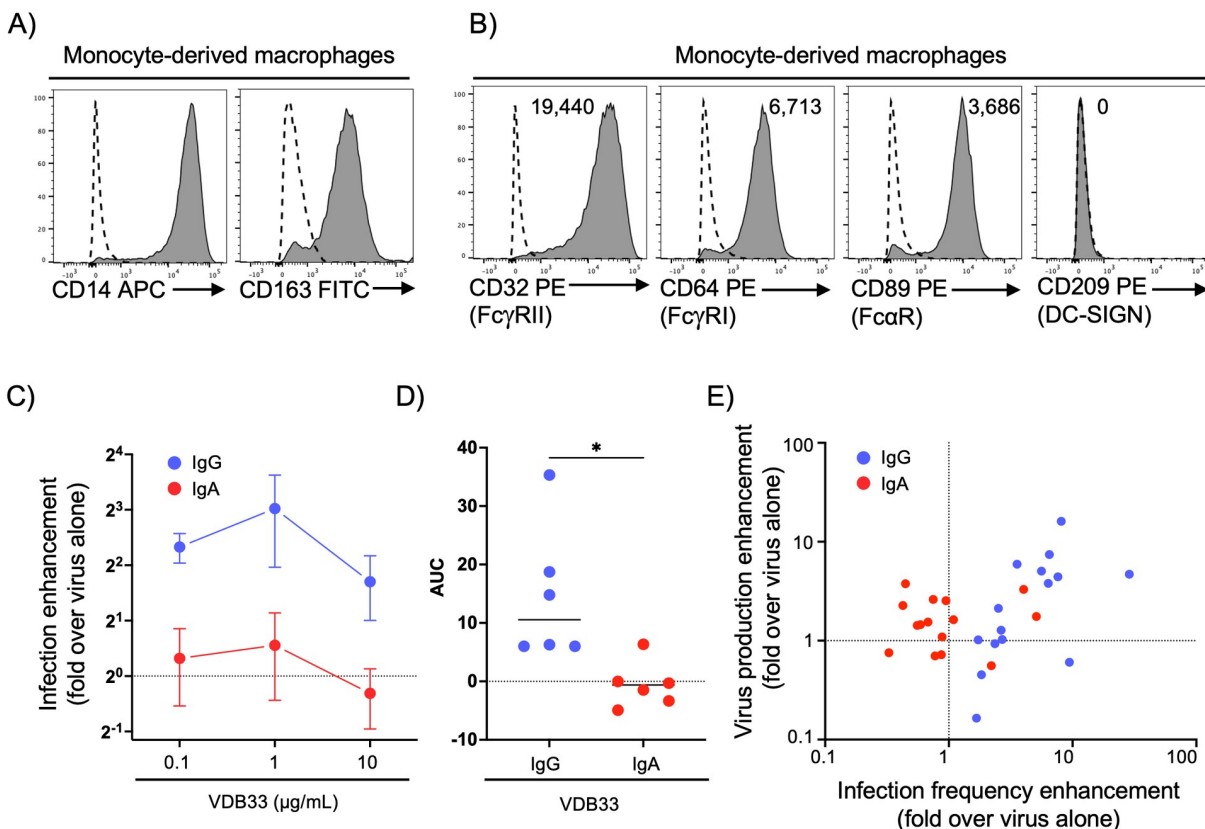

**Fig 2. DENV-specific IgG, but not IgA, mediates ADE in primary human monocyte-derived macrophages. A)** Assessment of CD14 and CD163 expression on primary human monocyte-derived macrophages by flow cytometry. Dashed line indicates isotype control. **B)** Quantification of FcγRIIa (CD32), FcγRI (CD64), FcαR (CD89), and DC-SIGN (CD209) on primary human monocyte-derived macrophages by flow cytometry. Histogram labels are isotype control-subtracted geometric mean fluorescence intensity (MFI) values. Dashed line indicates isotype control staining. **C)** Assessment of DENV infection-enhancing activity of DENV-specific IgG and IgA in primary human monocyte-derived macrophages. Data shown as fold change in primary human monocyte-derived macrophages cell infection frequency relative to DENV alone at the indicated antibody concentration. Dashed line indicates infection rate observed with virus alone (set to 1). **D)** The area under the ADE curves for each of 6 independent biological replicates of IgG and IgA infection experiments. **E)** Correlation between virus production as assessed by PCR and infected cell frequency human macrophage cell cultures infected with DENV alone or with IgG/ or IgA isotype mAbs. * p < 0.05, Wilcoxon matched-pairs test.

confirming the differentiated cells as bona fide macrophages (Fig 2B). Consistent with the literature, the rate of infection with DENV alone was low in our monocyte derived macrophages [37]. However, significant infection enhancement was observed with DENV complexed with IgG, but not IgA (Figs 2C, 2D, S5 and S6). The production of progeny DENV virions from infected macrophages was also enhanced by the addition of IgG proportionally to the frequency of DENV infected cells, while no such enhancement was observed in cultures treated with DENV/IgA immune complexes (Fig 2E). Thus, as was observed in the U937 cell line, DENV-reactive IgA exhibited no DENV infection-enhancing ability despite expression of FcαR on human monocyte-derived macrophages.

## DENV-specific IgG, but not IgA, enhances pro-inflammatory cytokine production by human macrophages

Although the exact immunopathologic mechanisms responsible for the development of severe dengue have not been determined, it is clear that serum levels of pro-inflammatory cytokines

and chemokines correlate with disease severity [8]. In light of the differential ability of DENV-specific IgG and IgA to enhance DENV infection in monocyte-derived macrophages, we hypothesized that DENV-specific IgG and IgA may also impact the cellular immunopathogenesis of DENV infection. To this end, we tested the supernatants of infected macrophage cultures with a multiplex cytokine assay to determine the levels of various cytokines associated with dengue disease [42].

The abundance of IFN-α2a (Figs 3A and S7) IFN-β (Figs 3B and S7), IL-6 (Figs 3C and S7), MIP-1α (Figs 3D and S7) and TNFα (Figs 3E and S7) in supernatants collected from human macrophage cultures infected with DENV alone or with DENV/IgG or DENV/IgA immune complexes was assessed using a multiplex cytokine array. These cytokines are well-established soluble markers of inflammation and several are known pyrogens, which is consistent with a pathologic contribution of IgG-mediated ADE to severe dengue. As we observed with the overall burden of DENV infected cells, DENV-specific IgG significantly enhanced the production of all these cytokines/chemokines above what was observed with DENV infection alone in a fashion directly proportional to the frequency of DENV infected cells in the culture. However, no such enhancement was observed upon the addition of DENV-specific IgA to the infection cultures. These results suggest that DENV-reactive IgA is capable of reducing both the DENV-infected cell burden relative to idiotype-matched IgG, and also limits infection-elicited cytokine production.

## IgA exhibits lower affinity for its cognate Fc receptor than IgG and does not facilitate low-valency binding to DENV-permissive cells

Having established the ability of DENV-reactive IgG, but not IgA, to mediate antibody-dependent enhancement of infection and consequent inflammation, we endeavored to determine the mechanism responsible. We have verified that the mAbs used in our experiments bind the DENV virion with equal affinity (S1 Fig) [28], indicating that this effect is not due to differential ability of DENV-specific IgG or IgA to bind DENV. We therefore tested the next possible source of difference, namely the binding of the IgG and IgA Fc domains to their cognate Fc receptors at a cellular and molecular level.

To this end, we first incubated U937 cells and monocyte-derived macrophages in the presence of human serum-derived polyclonal IgG or IgA followed by staining with the appropriate secondary antibody, and analyzed the total antibody-binding by flow cytometry. Strikingly, both U937 cells (Fig 4A) and human monocyte-derived macrophages (Fig 4B) bound more polyclonal IgG than polyclonal IgA at multiple antibody concentrations. To extend these observations to a more physiologically relevant setting, we performed a virus binding assay to assess the ability of DENV, DENV/IgG immune complexes, or DENV/IgA immune complexes to bind either U937 cells or human monocyte-derived macrophages. This assay was performed by incubating DENV, DENV/IgG, or DENV/IgA immune complexes with cells on ice to prevent virion internalization. After extensive washing, the abundance of cell-associated virions was assessed by RT-qPCR. Consistent with the relative receptor abundance and receptor/antibody affinity, DENV/IgG immune complexes bound both U937 cells and human monocyte-derived macrophages at a significantly higher rate than DENV alone (Fig 4C and 4D). However, IgA/DENV immune complexes exhibited no increased binding to these cells over that observed with DENV alone. These data are consistent with the patterns of infection burden described above, and support the conclusion that the differential affinity of IgG and IgA for their cognate Fc receptors is responsible for the variance in ability to mediate ADE.

While the observations described above might indicate that IgA/FcαR interaction are biologically extraneous in the setting of many infectious diseases, it must be noted that IgA is a

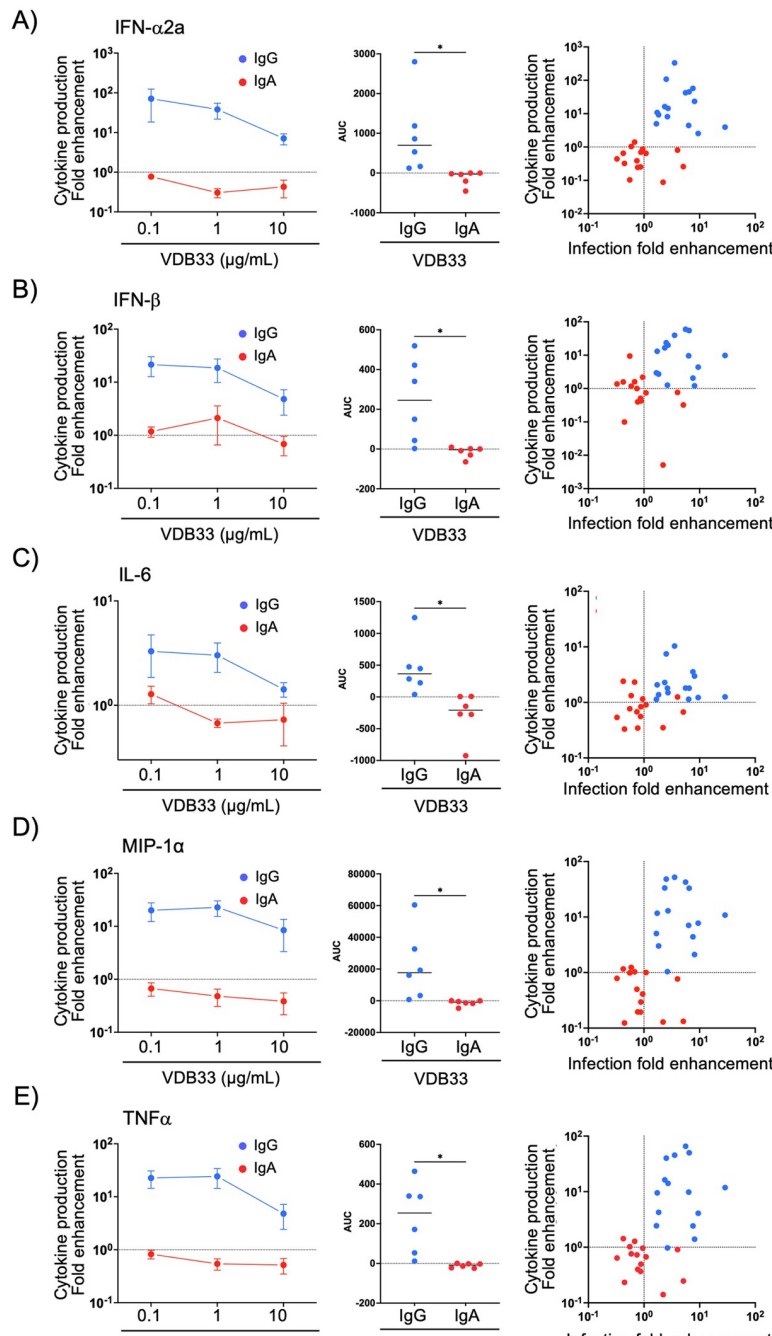

**Fig 3. IgG mediated antibody dependent enhancement increases pro-inflammatory cytokine production.**
Quantification of **A)** IFN-α2a, **B)** IFN-β, **C)** IL-6, **D)** MIP-1α, and **E)** TNFα in supernatants collected from human macrophage cultures infected with DENV alone or with DENV/IgG or DENV/IgA immune complexes. Data shown as fold change in cytokine production relative to DENV alone. Dashed line indicates cytokine production observed with virus alone (set to 1). Right panel indicates relationship between cytokine production enhancement (fold over DENV alone) and infection enhancement achieved at equimolar concentrations of IgG or IgA. * p < 0.05, Wilcoxon matched-pairs test.

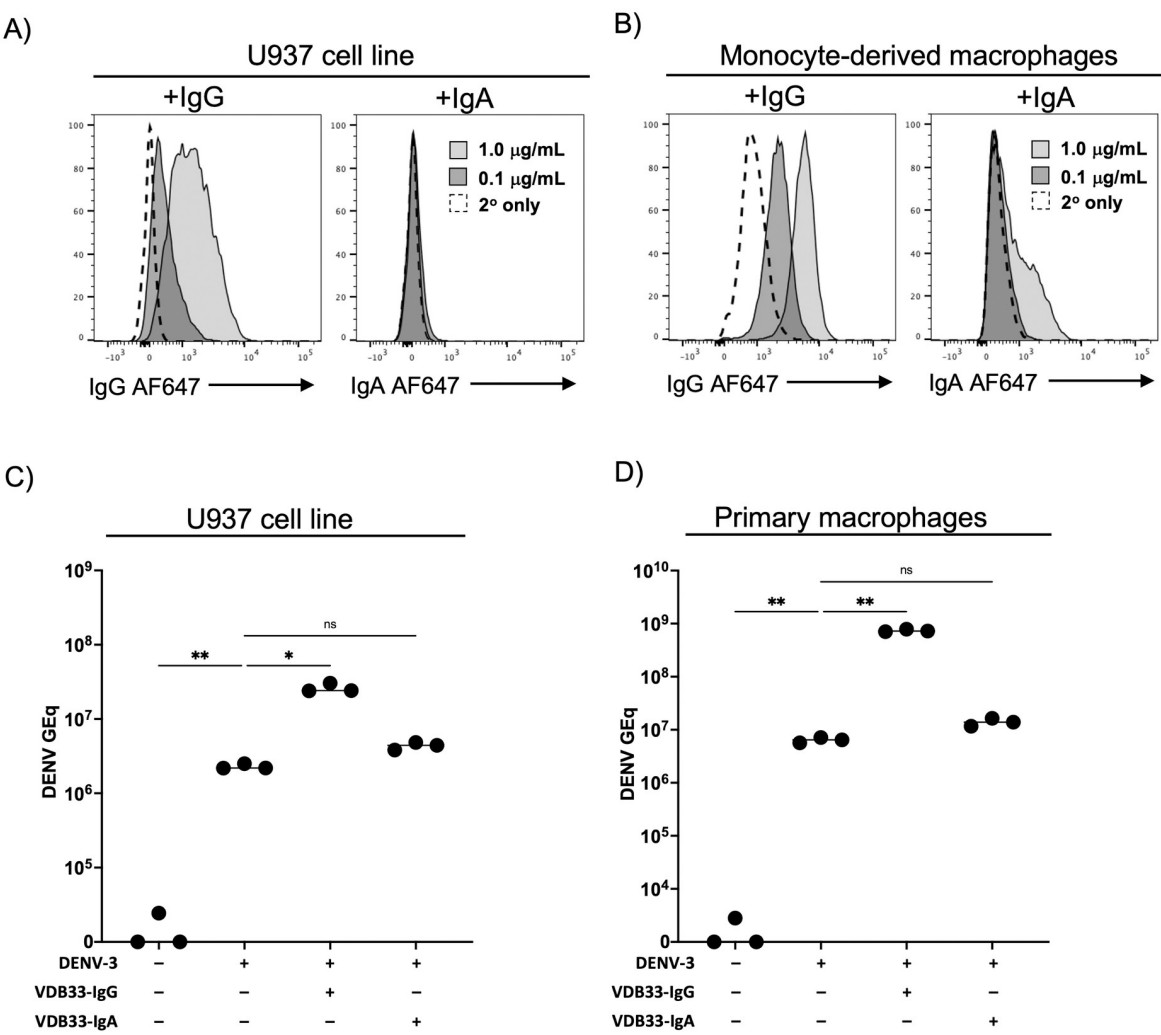

**Fig 4. IgG more efficiently binds its cognate Fc receptor than IgA. A)** Assessment of U937 cell ability to bind purified human serum-derived polyclonal IgG or IgA at the indicated concentration. Results are representative of 3 independent experiments. **B)**. Assessment of primary human monocyte-derived macrophages to bind purified human serum-derived polyclonal IgG or IgA at the indicated concentration. Results are representative of 3 independent experiments. **C)** DENV/IgG immune complexes bind FcγR and FcαR expressing cells more avidly than DENV/IgA immune complexes. DENV or DENV pre-complexed with VDB33-IgG or VDB33-IgA were added to U937 cells for 90 min at 4°C and extensively washed. Cell-bound virus quantified by RT-qPCR. Results are shown as background-subtracted GE/ml of cell lysate. **D)** DENV or DENV pre-complexed with VDB33-IgG or VDB33-IgA were added to monocyte-derived macrophages cells for 90 min at 4°C and extensively washed. Cell-bound virus quantified by RT-qPCR. Results are shown as background-subtracted GE/ml of cell lysate. * p < 0.05, ** p < 0.01, one-way ANOVA with Tukey post-hoc test.

potent opsonin which can facilitate the phagocytic uptake of bacteria, parasites, and even whole cells in an FcαR dependent fashion [30,43]. Indeed, we observed that the same DENV-reactive IgA mAb that exhibited no ADE activity in the assays described above was able to facilitate the phagocytic uptake of DENV-E protein coated polystyrene beads at levels only slightly lower than what was observed with IgG opsonized beads (S8 Fig). This leads us to suggest that FcαR expression and the affinity of IgA/FcαR interactions are tuned to selectively facilitate the uptake of high-valency antigens, while the higher expression and higher affinity of FcγR allows for the effective uptake of even small antigens such as IgG-opsonized virions.

## Fc receptor expression is dynamic during acute DENV infection

The data presented thus far suggest that IgA-mediated ADE is unlikely to occur due to the low affinity of IgA for its cognate Fc receptor and the relatively low expression of FcαR on monocytes, macrophages, and other permissive myeloid-linage cells. However, the expression of FcRs on monocytes, macrophages, and other cells permissive to DENV infection is highly dynamic and can be influenced by pro-inflammatory cytokines and other soluble inflammatory mediators [44,45]. While the cells chosen for our in vitro assays closely mirror the natural cellular reservoir of DENV in susceptible human hosts, our analysis leaves unaddressed the possibility that acute viral infection *in vivo* may modulate the expression of FcRs in such a fashion that would increase the possibility of IgA-mediated ADE.

Considering this, we sought to characterize the changes in FcR surface expression over the course of dengue infection to refine our understanding of the potential contribution of DENV-specific IgA to dengue pathogenesis. We selected pre-infection, acute illness, and convalescent PBMC samples collected from four previously flavivirus naïve patients infected with an attenuated DENV-3 challenge virus and assessed the expression of Fc-receptors FcγRIIa (CD32), FcγRI (CD64), and FcaR (CD89) on PBMC monocytes by flow cytometry. (S9 Fig). Consistent with previously published reports, CD14$^+$CD16$^-$ monocytes were the dominant subtype of observed at all time points, (Figs 5A, 5B, S10, S11 and S12), but the frequency of CD14$^+$CD16$^+$ intermediate monocytes increased during the peak of DENV infection and resolved by 28 days post infection (Fig 5A and 5B). Within the CD14$^+$CD16$^-$ classical monocyte compartment, expression of FcγRI increased significantly during the acute phase of DENV infection while the expression of FcγRIIa remained unchanged and the expression of FcαR was reduced relative to pre-infection levels (Fig 5A and 5C).

While these data suggest that DENV-elicited inflammation is unlikely to increase the susceptibility of circulating monocytes to IgA-mediated ADE during primary dengue by increased FcαR expression, ADE is a feature associated with heterologous secondary DENV infections. As the kinetics and composition of DENV-elicited inflammation and immunity differ in primary and secondary DENV infections [46], we next sought to assess FcαR expression in conventional monocytes during secondary DENV infections. To this end, we analyzed RNAseq data from a recent study of monocyte subsets in dengue, focusing the analysis on conventional monocytes sorted from healthy donors and patients experiencing acute secondary DENV infections 4–6 days after symptom onset (S13 Fig and S2 Table) [47]. Differential gene expression analysis revealed that FcγRI (CD64) gene expression was increased in monocytes derived from individuals experiencing secondary DENV infections relative to healthy controls, along with other canonical markers of acute viral infection such as IFI27, LY6E, SIGLEC1, and IFITM1 (Fig 5D and S3 Table). In contrast, no difference in FcαR (CD89) gene expression was observed between these groups (Fig 5D and 5E). In light of these analyses from both primary and secondary dengue, we posit that DENV-elicited inflammation may further reduce the likelihood of IgA-mediated ADE—but may increase the likelihood of IgG-mediated ADE due to the suppressed expression of FcaR and increased expression of FcγRs relative to monocytes at homeostasis.

## Discussion

In this study we have shown that DENV/IgG immune complexes, but not DENV/IgA complexes, can mediate enhancement of infection of DENV in both a monocytic cell line and in monocyte-derived macrophages. IgG-mediated ADE significantly increases production of pro-inflammatory cytokines including type I interferon, IL-6, MIP-1α, and TNF, which are associated with dengue symptoms and development of pathogenesis. Mechanistically, we have shown that this differential effect is largely attributable to lower avidity of IgA-opsonized

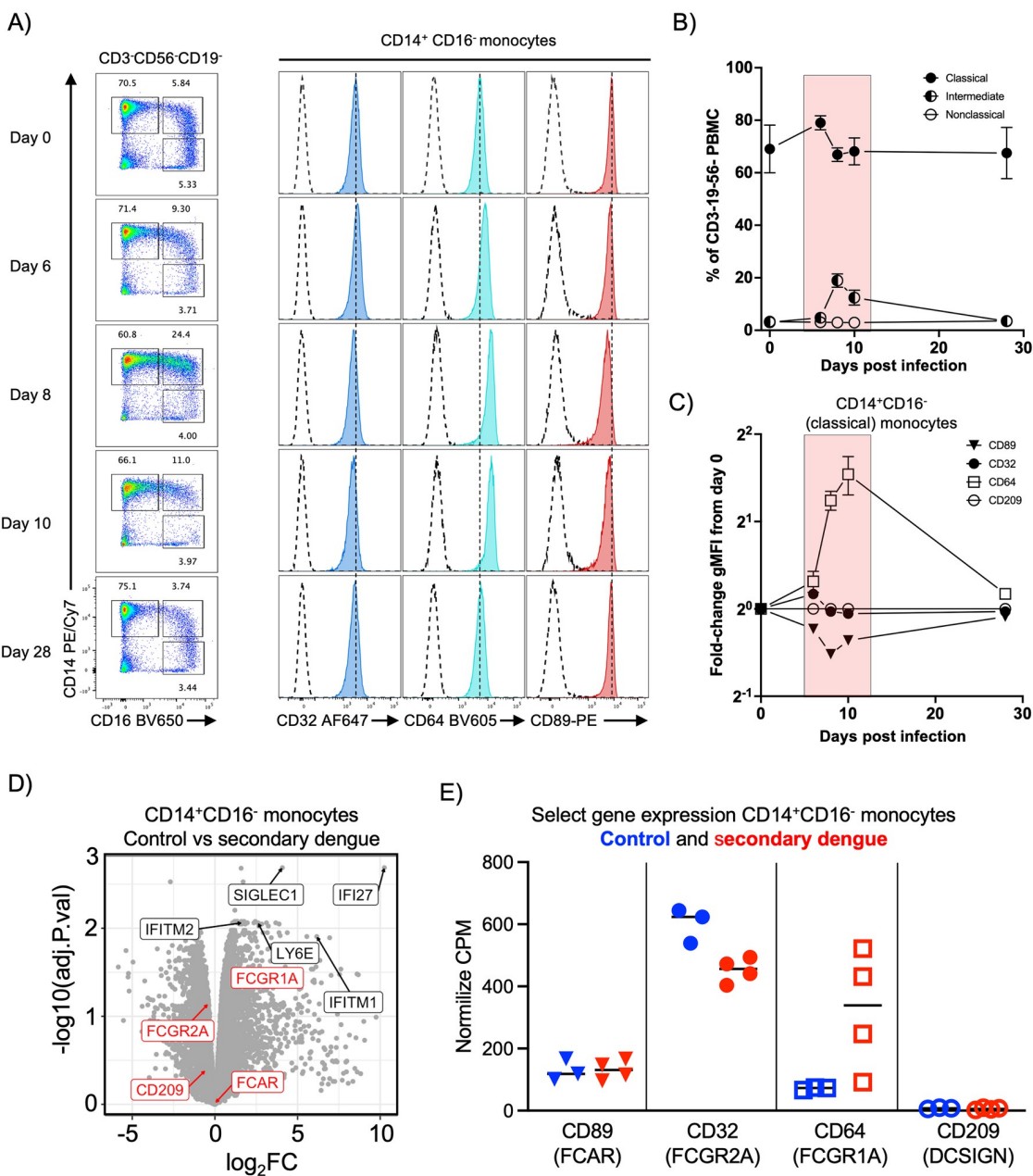

**Fig 5. FcαR expression in monocytes is not elevated by primary or secondary DENV infection.** Quantification of FcR expression on classical (CD14+CD16-) monocytes from individuals experiencing an experimental DENV-3 infection at the indicated time points. **A)** Representative flow cytometry plots showing distribution of CD14+CD16- (classical), CD14+CD16+ (intermediate), and CD14-CD16+ (non-classical) monocytes gated on viable CD3-CD56-CD19- leukocytes (left) and expression of select surface markers. **B)** Aggregate frequencies of CD14+CD16- (classical), CD14+CD16+ (intermediate), and CD14-CD16+ (non-classical) monocytes gated on viable CD3-CD56-CD19- leukocytes from 4 subjects. **C)** Aggregated fold-change of geometric mean fluorescence intensity of FcγRIIa (CD32), FcγRI (CD64), FcαR (CD89) and DC-SIGN (CD209) on CD14+CD16- monocytes. Stained samples were isotype control-subtracted prior to fold-change analysis. Red highlight indicates window of viremia following DENV-3 challenge. **D)** Differential gene expression analysis of sorted CD14+CD16- (classical) monocytes obtained from healthy donors and patients experiencing an acute secondary DENV infection 3–6 days after symptom onset. **E)** Expression of select genes from sorted CD14+CD16- monocytes obtained from healthy donors and patients experiencing an acute secondary DENV infection.

DENV for susceptible cells; both lower Fc—FcαR interaction affinity as well as lower expression of FcαR contribute to this effect.

Given that our infection studies with DENV virions revealed no evidence of ADE, we sought to confirm that our mAbs were capable of opsonizing targets for uptake by the FcαR. Adapting a bead-based phagocytosis assay [32,48], we showed that human monocytes were capable of uptaking both VDB33-IgG and -IgA opsonized polystyrene beads more efficiently than un-opsonized beads. These aligned with our other observations showing a clear efficiency advantage of IgG over IgA in opsonizing targets. These observations are consistent with the hypothesis that FcαR-dependent uptake is a function of the total *avidity* of the receptor-ligand interactions, wherein a larger quantity of IgA on the surface of a target could overcome the otherwise prohibitively low affinity of the individual interactions. This is consistent with the known ability of IgA to opsonize relatively large targets (such as bacteria and human cells) for phagocytic uptake by FcαR expressing cells, while at the same time exhibiting poor binding to the same receptors in monomeric or low-valency aggerates [30,43]. Indeed, previously reports have suggested that binding of IgA to FcαR only appears to occur when 5 or more IgA antibodies are multimerized in the same complex, with little or no binding of monomeric IgA [49]. This would also explain a similar observation by Tay and colleagues [48], who reported equal phagocytic efficiency of IgA and IgG opsonization of HIV env-conjugated beads, but greater efficiency of IgG opsonization of the HIV virion itself.

One implication of these data concerns the many attempts to define a correlate of protection for dengue [50], along with other studies of seroprevalence, such as those used as endpoint measures in vaccine trials. The data shown here, as well as our previously published work, present an often-unexamined factor that could additionally influence the probability of neutralization *versus* enhancement: the presence and quantity of DENV-reactive IgA in the serum. This is relevant because some measures of dengue seropositivity, such as HAI, could vary between samples as a function of DENV-reactive serum IgA titer, even if the IgM and IgG titers were otherwise identical by ELISA. If IgA were identified as a distinct factor conferring protective immunity, that in turn would suggest selective IgA deficiency, the most common primary immunodeficiency, as a potential risk factor for severe disease. Given its unique induction and decay kinetics compared to IgM and IgG, the IgA titer during dengue could also be potentially used to infer the timing of a prior infection serologically [21].

Another translational application of these data is the prophylactic and therapeutic potential of IgA mAbs. Given the risk of ADE, human monoclonal IgG antibodies proposed for dengue have included the LALA mutation to attenuate Fc region binding activity [51,52]. Another proposed neutralizing mAb initially characterized as a mouse mAb and suggested as a candidate for humanization [53] would presumably need a similar modification. However, several of the commonly used Fc domain modifications do not fully abrogate binding to FcRs or to C1q [54], while others show reduced FcRn binding, possibly leading to a reduced half-life *in vivo* [55]. By contrast, an IgA isotype therapeutic mAb would not require humanization or other modification to counter the risk of ADE. By the same reasoning, an IgA mAb could also be suitable as a prophylactic immunotherapy, especially for individuals who have already experienced a primary infection and whose window of potential re-exposure is known in advance (e.g., travelers, military personnel).

The majority of the *in vitro* cytokine production data presented herein are highly consistent with previously-published reports regarding the relationship between infection burden and cytokine production [56]. However, several elegant studies have suggested that ADE can suppress type I interferon production by human macrophages during DENV infection [37,57]. This is in contrast to the elevated IFN-α2a and IFN-β production observed in IgG-enhanced macrophage infection samples analyzed in our study and others [56]. While initially

contradictory, it must be noted that ADE-mediated suppression of type I interferon production was only observed at a narrow range of serum dilutions corresponding to the peak infection enhancement in this model, and that most other DENV-immune serum dilutions (including many that enhanced DENV infection) resulted in similar or significantly elevated rates of IFN production relative to DENV-infection alone [37]. This fact—coupled with other methodological differences in our studies including difference sources of infection-enhancing antibody (polyclonal serum vs purified mAbs) and the type of DENV used in the studies (DENV-2 vs DENV-3)–emphasizes the need for additional research to define the nature and prevalence of "cell-intrinsic" ADE in dengue.

There are several other limitations to our study which merit further investigation. While we used exclusively purified monoclonal IgA1 in our experiments, there is a small proportion (~10%) of dimeric IgA circulating in plasma [58], which according to some reports has a higher affinity for monocytes than monomeric IgA [59,60]. The IgA in our enhancement assays was all DENV-reactive monoclonal IgA, which would form immune complexes with DENV more readily than when diluted by non-DENV-reactive IgA. It has been shown that free IgA binding to the FcαR results in generally anti-inflammatory signaling, while immune complexed IgA binds more avidly and results in an activated, pro-inflammatory state [61–63]. This effect has been shown on a clinical scale, with infusion of pooled human serum IgA reducing inflammation in synovial infiltrates in rheumatoid arthritis as well as reducing symptoms of arthritis in human FcαR transgenic mice [64]. Given this, the anti-inflammatory contribution of free non-DENV-reactive IgA *in vivo* would further weigh against any inappropriate inflammation arising from IgA/DENV immune complexes. With the observation that IgA is incapable of mediating ADE in the systems described herein, we suggest that DENV-reactive IgA may possess the ability to not only limit DENV infection, but also DENV-associated immunopathogenesis.

## Methods

### Ethics statement

Studies involving human participants were reviewed and approved by the State University of New York Upstate Medical University (SUNY-UMU) and the Department of Defense's Human Research Protection Office. The participants provided their written informed consent to participate in this study.

### Viruses

DENV-3 (strain CH53489) stocks were prepared by propagating a low-passage inoculum in Vero cells. Supernatants were harvested, centrifuged at 3000g at 4˚ C to pellet suspended cells and debris, aliquoted, and stored at -80C. Infectious titer (PFU) was determined by plaque assay on Vero cells, with MOI of 3 was used in all subsequent infection assays.

### Cell lines

The U937 cell line was kindly supplied by Dr. Scott Blystone. Cells were maintained in RPMI media (Corning, #10-040-CV) supplemented to 10% (v/v) FBS, 1% penicillin/streptomycin, and 1% L-glutamine. Culture flasks were incubated in a humidified incubator at 37C, 5% $CO_2$.

### U937 ADE assay

*In vitro* antibody-dependent enhancement (ADE) of DENV-3 infection in U937 cells was quantified as previously described with slight modifications [28]. In brief, IgG or IgA mAbs

were incubated with virus (in sufficient amounts to infect 10%–15% of U937-DC-SIGN cells) at a 1:1 ratio for 1 h at 37˚C. This mixture was then added to a 96-well plate containing $5 \times 10^4$ U937 cells per well in duplicate. Cells were cultured for 48hr in a 37˚C, 5% CO2, humidified incubator, followed by flow cytometric analysis to determine infection frequency. FcR blocking in select assays was performed by the addition of anti-FcγRIIa/CD32 (Stemcell Technologies 60012 clone IV.3) and/or FcγRI/CD64 (Biolegend 305002 Clone 10.1) at a final concentration of 2ug/ml.

## Monocyte-derived macrophage differentiation

Monocytes from cryopreserved PBMC obtained from healthy normal donors were differentiated into macrophages largely following Boonnak et. al. [37]. In short, monocytes were isolated by negative selection according to the manufacture's protocol (Biolegend, #480059) and were resuspended in RPMI supplemented to with 100ng/mL M-CSF (Peprotech, # 300–25), 10% (v/v) FBS (Gibco), 1% penicillin/streptomycin, 1% L-glutamine. Cell density was adjusted to 1.25–1.5x$10^5$ cells per mL and 2mL/well plated in tissue culture-treated 24 well plates (Corning, #3524). Plates were centrifuged for 2 minutes at 500 g to adhere monocytes to the bottom of the well and incubated at 37˚C/5% $CO_2$. The day of isolation and plating was defined as day 1. The culture media was repleted on day 5 by adding 1mL of differentiation media per well.

## Monocyte-derived macrophage DENV infection

Inocula were prepared by mixing DENV with antibody dilutions (ADE conditions) or media (virus-only condition) and incubating at 37˚C/5% $CO_2$ for 60 minutes to allow for immune complex formation. Supernatants from 6-day monocyte-derived macrophage cultures were aspirated and 200 μL of inoculum added to each well. Plates were rocked to ensure even coverage by the inoculum and incubated for 2 hours at 37˚C/5% $CO_2$. After incubation, the inoculum was aspirated, wells were washed twice with 2mL RPMI medium, 500 μL differentiation media added, and plates incubated for 48 hours at 37C/5% $CO_2$. After 48 hours, the supernatant was removed, aliquoted, and stored at -80C for cytokine expression and viral burden analyses. Macrophages were detached from the plates by adding 600 μL of Accutase (Stemcell technologies, 07920) per well and incubated at 37˚C for 20 minutes, dissociated by pipetting, transferred to round-bottom polypropylene 96-well plates, and analyzed as described for each respective assay. Cytokine production from DENV infection macrophages was quantified using an MSD QuickPlex SQ120 instrument and a human U-PLEX cytokine panel (Meso Scale Diagnostics).

## Quantification of DENV infected cells

DENV-infected cell lines or macrophages were transferred to round-bottom polypropylene 96-well plates, washed, and fixed with IC Fixation Buffer (Invitrogen, 00-82222-49). Fixed cells were washed twice and permeabilized with 1X IC Permeabilization Buffer (Invitrogen, 00-8333-56). Cells were stained with the DENV PrM-reactive monoclonal 2H2 primary antibody (Millipore Sigma, MAB8705) at 1μg/mL, and with PE-goat anti-mouse IgG secondary antibody (#550589, BD Biosciences) at 0.4 μg/mL. The frequency of DENV infected cells was quantified on a BD LSRII flow cytometer (BD Biosciences). Data were analyzed with FlowJo version 10 (Becton Dickinson).

## Flow cytometry

Surface staining for flow cytometry analysis was performed in PBS supplemented with 2% FBS and Trustain FcX (Biolegend, 422301) at room temperature. Aqua Live/Dead (ThermoFisher,

L34957) was used to exclude dead cells in all PBMC phenotyping experiments. Antibodies and dilutions used for flow cytometry analysis are listed in S1 Table. Data collection was performed on a BD LSRII or Fortessa flow cytometer and analyzed using FlowJo v10.2 software (Becton Dickinson).

## Staining for surface-bound antibody

Macrophages or U937 cells were harvested, washed, resuspended in serum-purified human polyclonal IgG or IgA (I2511-10MG and I4036-1MG, Sigma Aldrich), and incubated at 4°C for 30 minutes. Cells were washed and stained with AF647-conjugated goat anti-human IgG or IgA as appropriate (2050–31 and 2040–31, Southern Biotech) and analyzed by flow cytometry as described above.

## Virus/antibody immune complex binding

DENV/antibody immune complexes were generated by mixing DENV-3 with VDB33-IgG or VDB33-IgA and incubating at 37C/5% $CO_2$ for 60 minutes and were then chilled to 4C. U937 cells or macrophages were harvested as described, counted, plated in a round-bottom polypropylene 96-well plate, centrifuged and supernatant decanted. Cells were resuspended in the chilled immune complexes, mixed gently by pipetting, and incubated at 4C for 90 minutes. After incubation, cells were washed 4 times in cold FACS buffer. After the last wash, cells were centrifuged, decanted, resuspended in 200mL RLT-plus buffer (Qiagen) with 10uL/mL 2-mercaptoethanol, and stored at -80C for analysis.

## Experimental DENV-3 human infection model

PBMC for flow cytometric analysis were obtained from a phase 1, open-label study (ClinicalTrials.gov identifier: NCT04298138) that was conducted between August 2020 and July 2021 at the State University of New York, Upstate Medical University (SUNY-UMU) in Syracuse, New York. Participants received a single subcutaneous inoculation of 0.7 x $10^3$ PFU (0.5ml of a 1.4 x $10^3$ PFU/mL solution) of the CH53489 DENV-3 infection strain virus manufactured at the WRAIR Pilot Bioproduction Facility, Silver Spring, MD (US FDA Investigational New Drug 19321). All participants were pre-screened to ensure an absence of preexisting flavivirus using the Euroimmun dengue, West Nile, and Zika IgG ELISA kits (Lübeck, Germany). Subjects were monitored in an outpatient setting unless the hospitalization criteria were met. The Dengue Human Infection Model and associated analysis was approved by the State University of New York Upstate Medical University (SUNY-UMU) and the Department of Defense's Human Research Protection Office.

## RNA sequencing gene expression analysis

Gene expression analysis was performed on previously published data [47] corresponding to sorted conventional monocytes isolated from three healthy individuals and four patients experiencing acute secondary DENV infections 3–6 days post symptom onset. FASTQs (S2 Table) were downloaded from NCBI GEO/SRA (series GSE176079) and mapped to the human reference transcriptome (Ensembl, Home sapiens, GRCh38) using Kallisto [65] version 0.46.2. Transcript-level counts and abundance data were imported and summarized in R (version 4.2.1) using the TxImport package (version 1.16.1) [66] and TMM normalized using the package EdgeR (version 3.30.3) [67,68]. Differential gene expression analysis performed using linear modeling and Bayesian statistics in the R package Limma (version 3.44.3) [69].

## Statistical analysis

All statistical analyses were performed using GraphPad Prism 9 (GraphPad Software, La Jolla, CA) with a p-value < 0.05 considered significant.

## Supporting information

**S1 Fig. Assessment of DENV1-4 E binding of VDB33-IgG and VDB33-IgA.**
(TIFF)

**S2 Fig. Gating strategy and representative flow plots for uninfected and antibody-enhanced infection of U937 cultures.**
(TIFF)

**S3 Fig. ADE curve in U937 cells using 4G2 staining.**
(TIFF)

**S4 Fig. Contribution of FcgRIIA and FcgRIA to IgG-mediated ADE in U937 cells.**
(TIFF)

**S5 Fig. Flow cytometry light micrograph analysis of thawed PBMC before and after macrophage differentiation.**
(TIFF)

**S6 Fig. Gating strategy and representative flow plots for uninfected and antibody-enhanced infection of macrophage cultures.**
(TIFF)

**S7 Fig. IgG mediated antibody dependent enhancement increases pro-inflammatory cytokine production.**
(TIFF)

**S8 Fig. IgG-more efficiently opsonizes polystyrene beads for phagocytosis than IgA.**
(TIFF)

**S9 Fig. Gating strategy for analysis of FcR expression on DHIM PBMC.**
(TIFF)

**S10 Fig. Expression of DC-SIGN (CD209) on live CD3-19-56-14+16- (classical) monocytes.**
(TIFF)

**S11 Fig. Evolution of intermediate monocyte FcR expression over the course of dengue infection.**
(TIFF)

**S12 Fig. Evolution of non-classical monocyte FcR expression over the course of dengue infection.**
(TIFF)

**S13 Fig. Analysis of RNAseq data obtained from sorted monocytes isolated from three (3) healthy individuals and four (4) individuals experiencing acute secondary DENV infections.**
(TIFF)

**S1 Table. Antibodies used for flow cytometry.**
(DOCX)

**S2 Table. Samples information for RNAseq analysis.**
(DOCX)

**S3 Table. Differential gene expression analysis of classical monocytes isolated from. health donors and donors experiencing a secondary DENV infection.**
(DOCX)

**S1 Data. Raw data for all manuscript figures.**
(ZIP)

## Acknowledgments

We gratefully acknowledge technical assistance provided by Lisa Phelps of the SUNY Upstate Medical University Flow Cytometry Core and the members of the Institute for Global Health and Translational Science (IGHTS) of SUNY Upstate Medical University.

## Author Contributions

**Conceptualization:** Adam D. Wegman, Adam T. Waickman.

**Formal analysis:** Adam D. Wegman, Mitchell J. Waldran, Kristen E. Baxter, Adam T. Waickman.

**Funding acquisition:** Adam T. Waickman.

**Investigation:** Adam D. Wegman, Mitchell J. Waldran, Lauren E. Bahr, Joseph Q. Lu, Kristen E. Baxter, Stephen J. Thomas, Adam T. Waickman.

**Resources:** Stephen J. Thomas, Adam T. Waickman.

**Writing – original draft:** Adam D. Wegman, Adam T. Waickman.

**Writing – review & editing:** Adam D. Wegman, Mitchell J. Waldran, Lauren E. Bahr, Joseph Q. Lu, Kristen E. Baxter, Stephen J. Thomas, Adam T. Waickman.

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
