## [Decision Letter · Decision Letter 0]

20 Jun 2023

Dear Dr Waickman,

Thank you very much for submitting your manuscript "DENV-specific IgA contributes protective and non-pathologic function during antibody-dependent enhancement of DENV infection" for consideration at PLOS Pathogens. As with all papers reviewed by the journal, your manuscript was reviewed by members of the editorial board and by several independent reviewers. In light of the reviews (below this email), we would like to invite the resubmission of a significantly-revised version that takes into account the reviewers' comments.

I am returning your manuscript with two reviews. The manuscript was well received by the reviewers and they both acknowledged the importance of this study to fill a knowledge gap for a role of IgA during ADE with dengue virus. However, major issues were identified, as you will see. After reading the reviews and evaluating the manuscript, I recommend Major Revisions. I am sorry I cannot be more positive, but we are looking forward to receiving your revised manuscript.

Please pay particular attention to the following reviewer suggestions and address them appropriately.

• Both reviewers raised major issues regarding the results in Figure 5. A main criticism was that the analysis was performed on cells isolated from individuals with a primary DENV-3 infection, however, ADE is more common during secondary infection. Please put the results in context of when ADE would be observed (based on timing and secondary DENV infection) and include additional data connecting the clinical observations with the enhanced infection observed in vitro.

• An additional method to evaluate enhanced DENV infection in your cell culture model will be needed.

• Please include additional discussion on the cytokine results under ADE conditions in relation to the literature.

We cannot make any decision about publication until we have seen the revised manuscript and your response to the reviewers' comments. Your revised manuscript is also likely to be sent to reviewers for further evaluation.

Sincerely,

Julie Fox, Ph.D.

Guest Editor

PLOS Pathogens

Sonja Best

Section Editor

PLOS Pathogens

Kasturi Haldar

Editor-in-Chief

PLOS Pathogens

orcid.org/0000-0001-5065-158X

Michael Malim

Editor-in-Chief

PLOS Pathogens

orcid.org/0000-0002-7699-2064

I am returning your manuscript with two reviews. The manuscript was well received by the reviewers and they both acknowledged the importance of this study to fill a knowledge gap in the role of IgA during ADE with dengue virus. However, both reviewers identified major issues, as you will see. After reading the reviews and evaluating the manuscript, I recommend Major Revisions. I am sorry I cannot be more positive, but we are looking forward to receiving your revised manuscript.

Please pay particular attention to the following reviewer suggestions and address them appropriately.

• Both reviewers raised major issues regarding the results in Figure 5. A main criticism was that the analysis was performed on cells isolated from individuals with a primary DENV-3 infection, however, ADE is more common during secondary infection. Please put the results in context of when ADE would be observed (based on timing and secondary DENV infection) and include additional data connecting the clinical observations with the enhanced infection observed in vitro.

• An additional method to evaluate enhanced DENV infection in your cell culture model will be needed.

• Please include additional discussion on the cytokine results under ADE conditions in relation to the literature.

Reviewer's Responses to Questions

**Part I - Summary**

Reviewer #1: This manuscript by Wegman and colleagues report an investigation into the the possibility of monomer IgA antibodies in mediating dengue virus (DENV) infection enhancement of myeloid derived cells via Fc alpha receptors. The authors compared the rate of virus uptake via this immune complex-Fc receptor interaction with that formed by IgG antibodies and Fc gamma receptors. They found that, unlike IgG-mediated uptake, IgA antibodies bound Fc alpha receptors with lower affinity and hence did not enhance infection of myeloid derived cells relative to DENV-only inoculation. They finally also showed that, during primary DENV-3 infection, the expression of FcgRI increased during the acute phase of symptomatic infection, whereas those of FcaR showed the opposite trend. They concluded that IgA antibodies against DENV have a protective role against ADE.

This is a well written manuscript and reports a finding that adds an important piece to the body of knowledge on dengue pathogenesis. The role of IgA antibodies have not been well studied and this group of investigators are leading the charge in filling this gap in knowledge.

Reviewer #2: This is a very clearly written, well designed and well executed study. It addresses an important knowledge gap of whether IgA plays a role in antibody dependent enhancement (ADE). ADE has been a key problem for the development of vaccines for dengue and it also complicates the potential therapeutic use of monoclonal antibodies in dengue. Furthermore, we still lack clear correlates of protection or immunopathology for dengue hence understanding the role of immune components including IgA is critical for vaccine design and development of specific therapies for dengue.

The data presented here is novel and of potential strong impact. This work shows that IgA, differently to IgG, is unable to mediate ADE in a myeloid cell line (U937) as well as in human in vitro monocyte-derived macrophages- with both cell types expressing FcgRI/IIa and FcaR. The authors go on to show that IgG-mediated DENV infection elicits production of pro-inflammatory cytokines by infected cells compared to infection with DENV alone or with DENV/IgA complexes and the production of these cytokines is proportional to the infection enhancement level. The authors also show that DENV-IgG complexes bind more efficiently to cells compared to DENV alone or DENV-IgA complexes –in line the increased infection rates of cells treated with DENV-IgG versus DENV alone. Lastly, using unique samples from a human controlled dengue infection model the authors show that following acute infection with an attenuated DENV3 strain, expression of FcgRI is upregulated while that of FcgR is downregulated in classical monocytes – further supporting the notion that IgA does not play a role in ADE.

My concerns are mainly around some of the experimental data (details in section below). Also, it would be useful if the authors could discuss their findings of increased cytokine production in ADE conditions in the context of published work. The authors show that cells infected with DENV-IgG complexes produce higher levels of type I IFNs compared to infection with DENV alone. However, the entry of DENV-Ab complexes through ADE was shown to suppress production of type I IFNs at peak ADE ab dilutions and to also induce IL-10 production (Boonnak et al JVI 2011). Studies also showed that ADE could down-regulate production of type I IFNs through mechanisms mediated my LILRB1 (Chan et al PNAS 2013).

**Part II – Major Issues: Key Experiments Required for Acceptance**

Reviewer #1: My only major concern is the data in Figure 5. ADE is not known to occur during primary infection and the changes in the expression of these receptors are occurring at a time when neither IgA nor IgG antibodies are expressed at high or even detectable levels. While it can be argued that a similar trend in the expression of these Fc receptors could be expected in secondary dengue, such an argument would ignore the known significant alterations in host response during ADE that can be detected as early as 24 hours from infection (Chan et al, mSphere 2019; 4:e00528-19). I do not think that omitting this set of data from the manuscript weakens the evidence that monomeric IgA antibodies are unlikely players in antibody-enhanced dengue and would recommend that the authors remove this data for a more focused paper.

Reviewer #2: 1. Fig 1 and 2: it is not clear how “infection rate” of U937 or monocyte-derived macrophages is measured (Fig 1B and 2C). Is this calculated based on the staining of DENV-infected cells with an anti-prM PE antibody as shown in Supplementary Figure 2 A-B (U937 showing approx. 10% infection) and Supplementary Fig 4 A-B (monocyte-derived macrophages showing 17% infection)? These anti prM ab staining profiles are not very convincing. Could the authors show staining using a different anti DENV ab? Can the authors show DENV virions released by the infected cell to prove that DENV is replicating as well as entering the cell?

2. Figure 5 shows that FCgRI is highly expressed in dengue infected individuals while FCgRII is not. Can the authors demonstrate in the experiments shown in Fig 1B and 1C that FcgRI is indeed playing a role in ADE and the effect is not only mediated by FcgRII (for example by using blocking abs to FcgRI/IIa)? This data would strengthen the validity of the in vitro data for the clinical context. Could the authors please comment on the timing of upregulation of FcgRI in patient monocytes, would this fit with the timing of when ADE is thought to occur during dengue infection?

**Part III – Minor Issues: Editorial and Data Presentation Modifications**

Reviewer #1: Lines 129-130: K562 is widely used but is not the gold standard for ADE assays. Primary monocytes, monocyte-derived macrophages/dendritic cells would make a more convincing gold standard.

Line 142 and elsewhere: Please specify the MOI used for all the experiments.

Figure 3: It would be useful to show the concentrations of these cytokines for each of the experimental conditions in the supplementary data. Such data would complement the fold changes shown in this Figure.

Figure 4c. Why is there a positive RT-qPCR finding for the no-DENV control? Perhaps the authors would consider showing the detection limit of the RT-qPCR used in this experiment?

Reviewer #2: 1. In Fig 5 it would be useful to show some flow cytometry plots for a representative donor to show the monocyte populations that the authors are looking at and how the markers of interest are expressed by classical monocytes for B. The authors could move this data now in supplem figs to the main figure.

2. While I agree that in vitro monocyte-derived macrophages are a good model for ex vivo macrophages, and I appreciate the difficult of working with ex vivo primary macrophages, in my opinion it is not correct to define monocyte-derived macrophages as “primary” cells as they are generated in vitro.

PLOS authors have the option to publish the peer review history of their article (what does this mean?). If published, this will include your full peer review and any attached files.

Reviewer #1: **Yes: **Eng Eong Ooi

Reviewer #2: No
---

## [Decision Letter · Decision Letter 1]

15 Aug 2023

Dear Dr Waickman,

We are pleased to inform you that your manuscript 'DENV-specific IgA contributes protective and non-pathologic function during antibody-dependent enhancement of DENV infection' has been provisionally accepted for publication in PLOS Pathogens.

Best regards,

Julie Fox, Ph.D.

Academic Editor

PLOS Pathogens

Sonja Best

Section Editor

PLOS Pathogens

Kasturi Haldar

Editor-in-Chief

PLOS Pathogens

orcid.org/0000-0001-5065-158X

Michael Malim

Editor-in-Chief

PLOS Pathogens

orcid.org/0000-0002-7699-2064

Reviewer Comments (if any, and for reference):

Reviewer's Responses to Questions

**Part I - Summary**

Reviewer #1: The authors have addressed all the concerns raised in their original manuscript. The revised version now reads very well and provides interesting insights into IgA antibody response and their impact on DENV infection.

Reviewer #2: (No Response)

**Part II – Major Issues: Key Experiments Required for Acceptance**

Reviewer #1: (No Response)

Reviewer #2: I am happy with the additional data provided by the authors which are convincing. I have no further concerns

**Part III – Minor Issues: Editorial and Data Presentation Modifications**

Reviewer #1: (No Response)

Reviewer #2: No concerns

PLOS authors have the option to publish the peer review history of their article (what does this mean?). If published, this will include your full peer review and any attached files.

Reviewer #1: No

Reviewer #2: No

---

## [Editor Report · Acceptance letter]

23 Aug 2023

Dear Dr Waickman,

We are delighted to inform you that your manuscript, "DENV-specific IgA contributes protective and non-pathologic function during antibody-dependent enhancement of DENV infection," has been formally accepted for publication in PLOS Pathogens.

Best regards,

Kasturi Haldar

Editor-in-Chief

PLOS Pathogens

orcid.org/0000-0001-5065-158X

Michael Malim

Editor-in-Chief

PLOS Pathogens

orcid.org/0000-0002-7699-2064